# Maternal Dexamethasone Exposure Induces Sex-Specific Changes in Histomorphology and Redox Homeostasis of Rat Placenta

**DOI:** 10.3390/ijms24010540

**Published:** 2022-12-29

**Authors:** Svetlana Trifunović, Branka Šošić Jurjević, Nataša Ristić, Nataša Nestorović, Branko Filipović, Ivana Stevanović, Vesna Begović-Kuprešanin, Milica Manojlović-Stojanoski

**Affiliations:** 1Institute for Biological Research “Siniša Stanković”—National Institute of Republic of Serbia, University of Belgrade, 11060 Belgrade, Serbia; 2Medical Faculty of Military Medical Academy, University of Defence, Institute of Medical Research, Military Medical Academy, 11000 Belgrade, Serbia; 3Clinic for Infectious and Tropical Diseases, Military Medical Academy, 11000 Belgrade, Serbia

**Keywords:** dexamethasone, pregnancy, rat, placenta, histology, oxidative stress

## Abstract

As the mediator between the mother and fetus, the placenta allows the most appropriate environment and optimal fetal growth. The placenta of one sex sometimes has a greater ability over the other to respond to and protect against possible maternal insults. Here, we characterized sex differences in the placenta’s morphological features and antioxidant status following dexamethasone (Dx) exposure. Pregnant rats were exposed to Dx or saline. The placenta was histologically and stereologically analyzed. The activity of the antioxidant enzymes, lipid peroxides (TBARS), superoxide anion and nitric oxide (NO) was measured. The decrease in placental zone volumes was more pronounced (*p* < 0.05) in female placentas. The volume density of PCNA-immunopositive nuclei was reduced (*p* < 0.05) in both sexes. The reduced (*p* < 0.05) antioxidant enzyme activities, enhanced TBARS and NO concentration indicate that Dx exposure triggered oxidative stress in the placenta of both fetal sexes, albeit stronger in the placenta of female fetuses. In conclusion, maternal Dx treatment reduced the size and volume of placental zones, altered placental histomorphology, decreased cell proliferation and triggered oxidative stress; however, the placentas of female fetuses exerted more significant responses to the treatment effects. The reduced placental size most probably reduced the transport of nutrients and oxygen, thus resulting in the reduced weight of fetuses, similar in both sexes. The lesser ability of the male placenta to detect and react to maternal exposure to environmental challenges may lead to long-standing health effects.

## 1. Introduction

The placenta is a temporary organ, acting during gestation as the primary exchange line for nutrients, gas, waste and communication between mother and fetus [1]. The chorioallantoic rat placenta morphologically has a discoid contour. It can be histologically divided into a maternal part/decidua and a fetal part consisting of a basal zone and a labyrinth zone [2]. Placental development and function are quite similar between female and male fetuses; however, after maternal exposure to stress hormones, hypoxia or poor diet, sex-specific differences in fetal development and long-term consequences (such as altered behavior, metabolic and cardiovascular disease) on offspring health become apparent during the life cycle [3,4]. In particular, males are more vulnerable to gestational complications and cardiovascular and neurodevelopmental disorders, and the placenta seems to play a role in the sex-specific relaying of the suboptimal maternal environment to fetuses [4,5].

Glucocorticoids (GC) control numerous processes required for efficient embryo implantation and the growth and development of the fetus and placenta [6]. Dexamethasone (Dx) is routinely administered when there is a risk of preterm delivery as a beneficial therapy for lung and cardiomyocyte maturation [6]. On the other hand, fetal exposure to increased concentrations of maternal GC modulates fetal development and has long-term consequences on the health of offspring and the onset of diseases in adulthood [7], the risk being higher in males than females [8]. For example, intrauterine growth restriction (IUGR) increased oxidative DNA damage, while redox function is enhanced in preeclampsia and IUGR [9]. We have recently demonstrated that maternal exposure to Dx induced adverse effects on the ovary of fetuses and offspring [10,11] and the fetal brain, which were more prominent in male fetuses [12]. Studies on sex-dependent changes in the placenta upon maternal Dx exposure are still insufficient. Keeping in mind the differences in structure between the human and rodent placenta as well as between sexes within the same species, the analysis of changes in particular placental zones needs focused and thorough examination [5].

In physiological pregnancy, the placenta produces reactive oxygen species (ROS) that are engaged in the progression of embryonic and fetal development, considering they participate in placentation and assist as signaling molecules, inducing the transcription of several genes. In contrast, the embryo and fetus have a low antioxidant capacity, and extra production of ROS would lead to a pro-oxidative state compromising fetal growth [13]; therefore, the fine balance is of tremendous importance. Measurements of oxidative stress indicators in maternal blood and urine demonstrate that pregnancy per se is a state of oxidative stress due to the high metabolic activity of the placenta and maternal metabolism adaptations [14,15]. The placenta is sensitive to oxidative stress, and ROS are known to provide a potent stimulus to apoptotic signaling [16]. In humans, glucocorticoid treatment of pregnant women induced sex-dependent placental changes, including alteration of placental oxidative stress [17]. Cuffe et al. [18] demonstrated that dexamethasone exposure increased TXNRD1 and GPX protein levels in male but not female placentas. To the best of our knowledge, no additional data thoroughly examines redox homeostasis and antioxidative enzyme activities in placentas upon Dx treatment in the context of sex difference. 

In the present study, we observed sex-specific changes in the morphological features and/or susceptibility of placentas to oxidative stress following maternal Dx exposure, which was used as a model of a challenging prenatal environment.

## 2. Results

### 2.1. Fetal and Placental Weight

Maternal Dx exposure between embryonic day (E) 16 and 18 resulted in significant (*p* < 0.05) fetal growth restriction and a reduction in the average weight of their placentas (Table 1) measured on E 21. For both sexes, the average fetal weight decreased (*p* < 0.05) by 16% and placental weight by more than 40% compared to the corresponding control values. The placental-to-fetal weight ratio was significantly lower after Dx exposure in comparison to corresponding controls.

### 2.2. Histological and Stereological Analysis

The regression of the basal zone (BZ) and full development of the labyrinth zone (LZ) were noted on E 21 in the placentas of all experimental groups (Figure 1A). Histological analysis revealed marked hypoplasia in the placentas of both male and female fetuses following Dx exposure.

In the control placentas, further histological examination confirmed the presence of thin decidua (D), BZ in regression and the most prominent LZ as expected for this embryonal age (Figure 2A). Spongiotrophoblasts occupied the largest part of the BZ (Figure 2A, arrow). Also, giant trophoblast cells were clearly visible with a large volume of cytoplasm and polyploid nuclei (Figure 2A, asterisk). Fetal vessels and maternal space were also visible in the LZ (Figure 2A, arrowhead). No visible histopathological changes could be distinguished in the placentas of male and female fetuses upon Dx exposure (Figure 2A(c,d)). 

Stereological measurements provided a more thorough determination of histological data. The absolute placental volume and the volume of the corresponding placental zones (decidua, basal zone and labyrinth zone) decreased following Dx exposure in comparison to the controls. After Dx exposure, the total volume decreased (*p* < 0.05) by 26% and 34% in the placentas of male and female fetuses, respectively. The sex difference was significant (*p* < 0.05), since the placentas of female fetuses had 12% lower (*p* < 0.05) total placental volume in comparison with the value obtained for the placenta of male fetuses (Figure 1B).

The absolute volumes of decidua, basal zone and labyrinth zone decreased (*p* < 0.05) by 22%, 14% and 30%, respectively, in the male placenta following Dx exposure. A more significant decrease was observed in the female placenta. The absolute volumes of decidua, basal zone and labyrinth zone decreased by 34%, 65% and 33%, respectively, as a consequence of Dx exposure (Figure 2B).

### 2.3. PCNA Immunostaining Analysis

The cell proliferation rate in the placentas was evaluated using proliferating cell nuclear antigen (PCNA) as an established marker of cell proliferation. PCNA immunoreactive nuclei were identified in the basal and labyrinth zones in the placentas of both sexes and controls. Following Dx exposure, the absence of PCNA-stained nuclei within the LZ was observed in both sexes (Figure 3A). The volume density of PCNA-immunostained nuclei in the placentas of the control male and female fetuses was 20.33 ± 1.32% and 19.7 ± 0.77%, respectively. This parameter was reduced after maternal Dx exposure in the placentas of male and female fetuses (*p* < 0.05) by 57% and 46%, respectively (Figure 3B). 

### 2.4. Oxidative Stress Analysis

The activity of major antioxidant enzymes involved in the placental antioxidative defense system decreased in the placentas of male and female fetuses following Dx exposure (Figure 4). Namely, the catalase (CAT) activity was reduced (*p* < 0.05) by 12% in the placentas of both male and female fetuses after Dx exposure (Figure 4A). The superoxide dismutase (tSOD) activity was reduced (*p* < 0.05) by 40% in the placentas of male fetuses and by 22% in the placentas of female fetuses (Figure 4B). The sex difference was significant (*p* < 0.05) in the case of this parameter (Figure 4B). The glutathione (GSH) value decreased (*p* < 0.05) by 40% in the placentas of male fetuses and by 30% in the placentas of female fetuses following Dx exposure (Figure 4C). The sex difference was significant (*p* < 0.05) in the case of this parameter both in the control and Dx-exposed placentas. Also, Dx exposure triggered a decrease in glutathione peroxidase (Gpx) activity (*p* < 0.05) by 69% in the placentas of male fetuses and by 70% in the placentas of female fetuses (Figure 4D). The thiobarbituric acid reactive substances (TBARS) values, as an indicator of the lipid peroxidation process, were significantly increased (*p* < 0.05) by 100% and 60% in the placentas of male and female fetuses, respectively, following Dx exposure (Figure 4E). The sex difference in the TBARS value was noticed in the controls (*p* < 0.05) as well as after the Dx exposure (*p* < 0.05) (Figure 4E). The concentration of superoxide anion radical (O_2_^•−^) was 11% lower (*p* < 0.05) only in the placentas of female fetuses upon Dx exposure (Figure 4F). Following Dx exposure, the NO concentration increased (*p* < 0.001) by 106% in the placentas of male fetuses and by 110% in the placentas of female fetuses (Figure 4G). Two-way factorial ANOVA testing of the effects of sex and Dx treatment on CAT, tSOD GSH, Gpx, TBARS, O_2_^•−^ and NO are summarized in Table 2. The result of Dx treatment has a more significant impact on the examined parameters than the fetal sex (Table 2).

## 3. Discussion

Although sex-specific differences are recognized in fetal responses to antenatal Dx exposure, few studies have approached the importance of fetal sex in placental development, which has a role during pregnancy and influences offspring programming. Using our previously well-characterized animal model of Dx exposure in rats [11,12], the current study aimed to investigate its putative sex-specific effects on placental histomorphology and redox homeostasis. 

The results of this study clearly confirmed an association between maternal Dx exposure and decreased fetal growth in line with the results of our [11] as well as other researchers’ studies [19]. When analyzing the difference between female and male fetuses, it was revealed that maternal Dx exposure decreased the absolute fetal and placental weights and the placenta-to-fetal body weight ratio with no statistical difference between the sexes. On the contrary, a previous study [20] demonstrated that exposure to corticosterone had no impact on mouse fetal weight and increased placenta weight only in male fetuses. In contrast to corticosterone, which is highly metabolized by endogenous placental 11β-hydroxysteroid dehydrogenase enzyme type 2, Dx may easily cross the placental barrier [21], thus exerting more profound effects on fetuses. However, another study revealed a decreased weight of the placenta only in female fetuses after Dx exposure [22]. In their work, Cuffe et al. [18] demonstrated a sex-specific difference in the expression of glucocorticoid receptors in the mouse placenta in line with the more profound effects of glucocorticoid treatment on female placentas. 

Stereological analysis showed a decrease in the absolute volume of the placenta following Dx exposure on E 21 in both sexes, which is in correlation with the observed decline in the PCNA expression and volume density estimation. More pronounced decreases in decidua, basal and labyrinth absolute zone volumes were observed in the placentas of female fetuses. During the rat gestation period, the placental structure changes dramatically. The decidua and basal zone undergo regression after E 15, while the labyrinth zone develops to the greatest extent till E 21 [23]. Considering that the LZ forms the majority part of the placenta on E 21 and its role in the maternal–fetus O_2_/CO_2_ exchange, providing nutrients to the fetus and removing waste products, the observed hypoplasia of the LZ and decreased cell proliferation, triggered by its vulnerability to Dx, were highly correlated with fetal and placental growth retardation [24]. 

The Dx action is exclusively mediated by the glucocorticoid receptor (GR), which acts as a transcription factor and has at least five characterized splice variants (GRα, GRβ, GRγ, GRA and GRP). Each of these splice variants has eight different initiation sites, resulting in up to 40 potential GR isoforms [25,26,27]. It is important to point out that different GR receptor isoforms regulate different downstream signaling pathways [28,29]. Male and female placentas from mice have different GR isoform profiles in line with the greater sensitivity of female placentas to Dx: the nuclear expression of the anti-apoptotic GR isoform is higher in placentas of males than in female fetuses, while nuclear expression of the pro-apoptotic GR isoform is higher in female placentas [18]. Therefore, it seems logical to assume that a more significant decrease in the placenta and placental zone volumes in females is due to the differential expression of GR isoforms and enhanced sensitivity of females to glucocorticoids compared to males. Further studies of GR expressions in our model are needed to confirm this assumption.

Pregnancy per se is a state of oxidative stress, i.e., an imbalance of ROS production, which damages lipids, proteins, carbohydrates and nucleic acids and the ability of antioxidant defenses to scavenge them [30,31]. The most important antioxidant enzymes, which are SOD, catalase, glutathione, glutathione peroxidase and glutathione S-transferase, are present in the rat placenta [32]. In physiological pregnancies, oxidative stress is eliminated by the upregulation of antioxidant defenses neutralizing ROS and reactive nitrogen species. Maternal Dx exposure triggered oxidative and nitrosative stress in the placentas of both fetal sexes, which provoked a decline in antioxidant defense capacity. Aside from the direct effect on antioxidative enzyme activities, the obtained results may be due to Dx-induced upregulation of gene and protein expressions for those enzymes under our experimental conditions. Previous studies demonstrated that maternal Dx exposure affects gene and protein expression of oxidative stress markers [16,18]. Namely, markers of placental damage were positively correlated, while defense system markers were negatively correlated with Dx exposure. The obtained results indicate that the effect of Dx treatment had a more significant impact on the examined parameters than the fetal sex, even though female placentas exerted greater changes in comparison to males.

SOD is an essential line of the enzymatic defense system since it catalyzes the dismutation of O_2_^•–^ in H_2_O_2_ and O_2_ [33], while CAT and GPx are responsible for converting H_2_O_2_ into water and O_2_ mainly when high levels of H_2_O_2_ are found [34]. The reduction in SOD and CAT activity was found in the placentas of both male and female fetuses upon exposure to Dx without apparent sex differences. In contrast, lower GPx activity was identified in the placentas of both sexes yet more prominently in the female placentas. GSH levels were also reduced upon exposure to Dx in a sex-specific manner, and higher concentrations of TBARS were found in the placentas of female fetuses in the controls and Dx-exposed placentas. An increase in NO concentration in the placentas of both sexes and a decrease in the oxidative stress marker O_2_^•−^ only in the placentas of female fetuses were observed following Dx exposure. In vitro analyses showed that Dx exposure generates ROS overproduction by regulating genes involved in ROS generation via glucocorticoid receptor activation [35]. Apoptosis is also known to be initiated by oxidative stress [36], which may be one of the suspected mechanisms of the histomorphological changes detected under our experimental conditions. Moreover, in line with Cuffe et al., the sex difference in the intensity of oxidative stress between male and female placentas can be, at least partly, also due to sex-specific alterations in GR expression [18].

Reduced placental size means a reduction in the surface area for the transport of nutrients and oxygen to the fetus [37], which under our experimental conditions probably caused the decreased body weight of the fetuses. This is in line with the results of other researchers who reported a positive correlation of placental size to body weight at birth term in various species [38]. Poor/excess nutrition, obstetric complications (preeclampsia, gestational diabetes and hypertension) or abuse of toxic substances were also associated with decreased fetal growth, leading to low birth weight due to alterations in placental function [38,39,40,41]. Dx exposure decreased the expression of genes involved in cell division [42] and induced oxidative stress [16] in a murine-late placenta. Thus, both mechanisms that we examined, i.e., reduction in the proliferation rate, demonstrated a reduced number of PCNA-immunopositive nuclei as well as increased oxidative stress, which may directly contribute to the demonstrated effects of dexamethasone on the reduction in placental size and possibly its transport function, consequently resulting in lower fetal weight. 

There was no significant difference in the fetal weight between sexes despite the fact that the female placentas were more affected by the treatment. Namely, the decrease in placental volume, the volume of all placental zones and the activity of some antioxidative protection enzymes was more prominent in the placentas of female fetuses. It is crucial to note that placental growth restriction and function can cause a decreased blood supply to the fetus with long-lasting effects. Aside from physical development, this may adversely compromise their reproductive and hormonal systems and cause cancer and neurodegenerative disorders during life. Our previously published results under the same experimental conditions demonstrated adverse effects of antenatal Dx treatment on the ovaries of fetuses and offspring [10,11]. On the other hand, the impact on the fetal brain was more clearly evident in male fetuses [12] in line with the results of other researchers [4,5]. However, Huang et al. [43] reported significant alterations in the glutaminergic system of the hippocampus in female offspring. Therefore, it seems logical to speculate that the hereby demonstrated changes in placentas contributed directly to the observed immediate adverse effects on fetal development and the long-term impact evident in offspring of both sexes [3,4]. 

In conclusion, this study provided original evidence that maternal exposure to Dx directly affected the placenta: impaired its weight and volume, the volume of all placental zones as well as cell proliferation rate and elicited oxidative stress in a sex-specific manner. The obtained changes most probably reduced placental efficiency for transport of nutrients and oxygen in line with reduced fetal body weight. Some of the observed changes were more prominent in the placentas of female fetuses, but the effect of Dx treatment had a greater impact than the fetal sex. 

## 4. Materials and Methods

The experiments were performed on naturally cycling female Wistar rats kept in the facilities of the Institute for Biological Research “Siniša Stanković”, Belgrade, Serbia, under standard environmental conditions (a 12 h light/dark cycle, 22 ± 2 °C). The females’ weight was at least 220 g before entering the protocol. Females showing a regular 4-day estrus cycle were included in the experimental procedure. The first confirmation of pregnancy was the presence of sperm in vaginal smears during proestrus after females were caged with a fertile male overnight. The day was considered as day 0 of gestation. Gravid females were divided into two groups. The first group, dexamethasone exposure females (*n* = 10), were injected subcutaneously with 0.5 mg/kg dexamethasone phosphate—Dx (Krka, Novo Mesto, Slovenia, dissolved in 0.9% saline) during embryonic days 16, 17 and 18. The second group, the control group (C), was injected with an equal volume of saline in the same manner. The final average weight of pregnant females on day 21 of gestation was 347.2 ± 24.1 and 327.5 ± 28.7 in the control and Dx group, respectively. Fetuses and placentas in each group were obtained by cesarean section on day 21 of gestation. Sex was distinguished based on the male’s larger genital papilla and its longer distance from the anus. Placentas were collected as follows: one placenta from a male or female fetus was used per litter (*n* = 6 male/female fetuses from 8–10 L per group) for histomorphology/stereology (put in fixative–paraformaldehyde) and oxidative status analyses (stored at −20 °C).

### 4.1. Histomorphology and Stereology Analyses

After 48 h in 4% paraformaldehyde, the placentas were routinely processed into paraffin blocks. Serial sections of the placentas (5 µm thick) were obtained with a rotary microtome (RM 2125RT, Leica Microsystems, Gmb, Solms, Germany). Further, placentas were routinely stained with hematoxylin or stained immunohistochemically against PCNA. 

All stereological analyses were carried out using a workstation comprising a microscope (Olympus, BX-5; Olympus Microscopy, Tokyo, Japan) equipped with a microcator (Heidenhain MT1201, Heidenhain, USA) to control movements in the z-direction (0.2 μm accuracy), a motorized stage (Prior) for stepwise displacement in x−y directions (1 μm accuracy) and a CCD video camera connected to a 19’ PC monitor. The whole system was controlled by the newCAST stereological software package. The stereological analysis included the absolute volume of the placenta and absolute volumes of placental zones: decidua, basal zone and labyrinth zone. The volumes were estimated according to Cavalieri’s principle [44,45]. The sampling of hematoxylin-stained placentas was systematically uniform from the random start (every 40th section from each tissue block was analyzed). Mean section thickness was estimated using the block advance (BA) method [46], and we found no variation from 5.0 μm as set in the microtome. The total volume (mm^3^) of the placentas was determined by multiplying the sum of the areas by the interval between the sections (200 μm) according to the formula:$$\bar{V}=a(p)\cdot BA\cdot \sum_{i=1}^{n}Pi$$
where *a*(*p*) is the area associated with each sampling point, *BA* (block advance) is the mean distance between two consecutively studied sections (real section thickness 5 μm × 40) and Σ*Pi* is the sum of points hitting a given target.

Immunolocalization of the proliferating cell nuclear antigen—PCNA. After dewaxing, hydration and rinsing in 0.01 M phosphate-buffered saline (PBS; pH 7.6 for 10 min), the sections were exposed to microwaves (700 W) in 0.05 M citrate-buffered saline (pH 6.0; for 2 × 10 min) for antigen retrieval. Subsequently, the sections were incubated (15 min) in a hydrogen peroxide solution in methanol to block endogenous peroxidase and washed in PBS (10 min). The sections were preincubated in normal goat serum (1:10) for 30 min and then incubated overnight with anti-PCNA (1:5000; ab29, Abcam) to block nonspecific staining. PCNA is a marker for the cell cycle; its expression peaks in the late G1 and S phases of the cell cycle [47]. After washing in PBS, the sections were incubated for another 1 h with the secondary antibody (polyclonal goat-anti-mouse; 1:200; ab6789, Abcam) and again rinsed with PBS. The 0.05% 3,3-diaminobenzidine tetrachloride liquid substrate chromogen system was used for antibody localization. Control sections were incubated with rabbit nonimmune serum at the same concentration as the primary antibody (in omission of the primary antibody), which resulted in the complete loss of immunoreactivity in the placenta section.

The percentages of PCNA-stained nuclei were obtained using five immunostained sections from five levels of the placenta for each placenta. The volume density or percentages of immunolabeled PCNA nuclei were calculated by counting the points hitting the PCNA (brown) stained nuclei and dividing these with the points hitting non-stained placental tissue areas × 100.

### 4.2. Determination of Oxidative/Nitrosative Status Parameters

The whole procedure of tissue preparation was performed on ice. Around 100 mg of placental tissue was transferred into 1 mL ice-cold buffered sucrose (0.25 mol/L sucrose, 0.1 mmol/L EDTA in sodium–potassium phosphate buffer, pH 7.2). Aliquots (1 mL) were placed into a glass tube homogenizer (Tehnica Manufacturing, Zelezniki, Slovenia). Homogenization was performed twice with a Teflon pestle at 800 rpm (1000× *g*) for 15 min at 4 °C. The homogenates were centrifuged at 2500× *g* for 30 min at 4 °C. The supernatants were sonicated by three cycles (30 s sonication and 5 s pause) and used for analysis.

The parameters of oxidative/nitrosative stress as well as antioxidative capacity were measured in the placenta of male and female rats. The protein content in the supernatant was measured by the method of Lowry using bovine serum albumin as the standard [48]. 

Malondialdehyde (MDA) was determined by the spectrophotometric method of Villacara and coworkers [49]. MDA, a secondary lipid peroxidation product, gives a red-colored pigment following incubation with thiobarbituric acid-TBA reagent (15% trichloroacetic acid and 0.375% TBA, water solution) at 95 °C at pH 3.5. The absorbance was measured at 532 nm. The results were expressed as μmol MDA per mg of proteins. 

After deproteinization, the production of NO was evaluated by measuring the nitrite and nitrate concentrations (NO_2_ + NO_3_). The nitrites were assayed directly spectrophotometrically at 543 nm using the colorimetric method of Griess (Griess reagent: 1.5% sulfanilamide in 1M HCl plus 0.15% N-(1-naphthyl) ethylenediamine dihydrochloride in distilled water). However, the nitrates were earlier converted into nitrites by cadmium reduction [50]. The results were expressed as mmol per mg proteins.

The content of superoxide anion radical (O_2_^•−^) was quantified by the method based on the reduction of nitroblue-tetrazolium (NBT) to monoformazan by O_2_^•−−^ in the alkaline nitrogen saturated medium. The yellow color of the reduced product was measured spectrophotometrically at 550 nm [51]. The results were expressed as μmol of reduced NBT/min/mg proteins.

Total superoxide dismutase activity (tSOD) was measured spectrophotometrically as the inhibition of spontaneous auto-oxidation of epinephrine at 480 nm. The kinetics of sample enzyme activity was followed in a carbonate buffer (50 mM, pH 10.2, containing 0.1 mM EDTA) after adding 10 mM epinephrine [52]. The results were expressed as U SOD per mg proteins.

Catalase activity (CAT) was estimated by the spectrophotometric method. Ammonium molybdate forms a yellow complex with H_2_O_2_ and is suitable for measuring both serum and CAT activity in the tissue [53]. Kinetic analysis was performed at 405 nm. Data were expressed as mU CAT per mg of proteins.

The method of estimating glutathione peroxidase (GPx) activity is based on an indirect determination of GPx activity by the spectrophotometric measurement of NADPH consumption at 340 nm. For a moment, GPx catalyzes the reduction of (lipid) hydroperoxides to (alcohols)/H_2_O using reducing equivalents of GSH, which subsequently becomes oxidized. Additionally, regeneration of depleted GSH occurs throughout the reduction of GSSG to GSH, catalyzed by the enzyme glutathione reductase, which utilizes NADPH as a donor of reducing equivalents. Reduction of every mole of GSSG requires one mole of NADPH [54]. The results were expressed as U GPx per mg of proteins. 

Total glutathione content (GSH+1/2 GSSG, in GSH equivalents) was determined with the DTNB–GSSG reductase recycling assay. The rate of formation of 5-thio-2-nitrobenzoic acid (TNBA), which is proportional to the total glutathione concentration, was followed up spectrophotometrically at 412 nm [55]. The results of GSH were expressed as nmol GSH per mg of proteins. 

### 4.3. Reagents

All chemicals were of analytical grade. The following compounds were used in this study: glutathione reductase (EC 1.6.4.2), Type III, from yeast [9001-48-3], Sigma Chemical Co (St. Louis, MO, USA)—highly refined suspension in 3.6 M (NH_4_)_2_SO_4_ at pH 7.0; 2500 U/1.6 mL (9.2 mg prot/mL—biuret) 170 U/mg proteins (Note: 1 unit reduces 1 μmol GSSG/min, pH 7.6 at 25 °C); saline solution (0.9% *w*/*v*) (Hospital Pharmacy, Military Medical Academy, Belgrade, Serbia); glutathione, glutathione disulfide and NADPH (Boehringer Corp.—London, UK); sodium nitrate (NaNO_3_) (Mallinckrodt Chemical Works—St. Louis, MO, USA); ethylenediaminetetraacetic acid—EDTA, nitrobluetetrazolium—NBT, N-(1-Naphthyl)ethylenediamine dihydrochloride, sulfanilamide, amonium molybdare, potassium sodium tartarate tetrahydrate, sodium hydroxyde, gelatine, 5,5′-dithiobis(2-nitrobenzoic acid)—DTNB, epinephrine (Sigma-Aldrich—St. Louis, MO, USA); sodium phosphate—Na_2_HPO_4_, potassium dihydrogen phosphate—KH_2_PO_4_, trichloroacetic acid, thiobarbituric acid (Merck—Darmstadt, Germany). Deionized water was prepared by the Millipore Milli-Q water purification system (Waters—Millipore, Milford, MA, USA). All drug solutions were prepared on the day of the experiment.

### 4.4. Statistical Analysis

Statistical analyses were performed using the statistical package program Statistica 10. All results are presented as mean ± SD. For stereological and oxidative stress results, *n* = 6 per group, while for fetal and placental weights, *n* = 15 per group. The significance of the difference between groups was analyzed using two-way ANOVA followed by the Bonferroni post hoc test. A confidence level of *p* < 0.05 was considered statistically significant. 

## Figures and Tables

**Figure 1 ijms-24-00540-f001:**
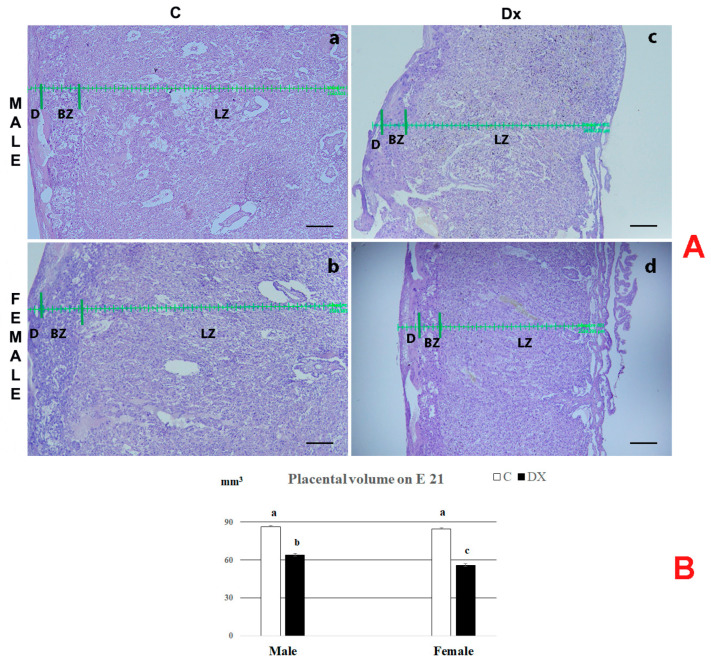
(**A**) Hematoxylin-stained sections of placentas from control (C; (**a**,**b**)) and dexamethasone-exposed (Dx; (**c**,**d**)) male and female fetuses on E 21. Decidua (D), basal zone (BZ), labyrinth zone (LZ), scale bar = 300 μm; (**B**) Absolute placental volume (mm^3^). Data are presented as mean ± SD, *n* = 6 placentas per group. Results were analyzed by two-way ANOVA. The groups not sharing a joint letter are significantly different (*p* < 0.05).

**Figure 2 ijms-24-00540-f002:**
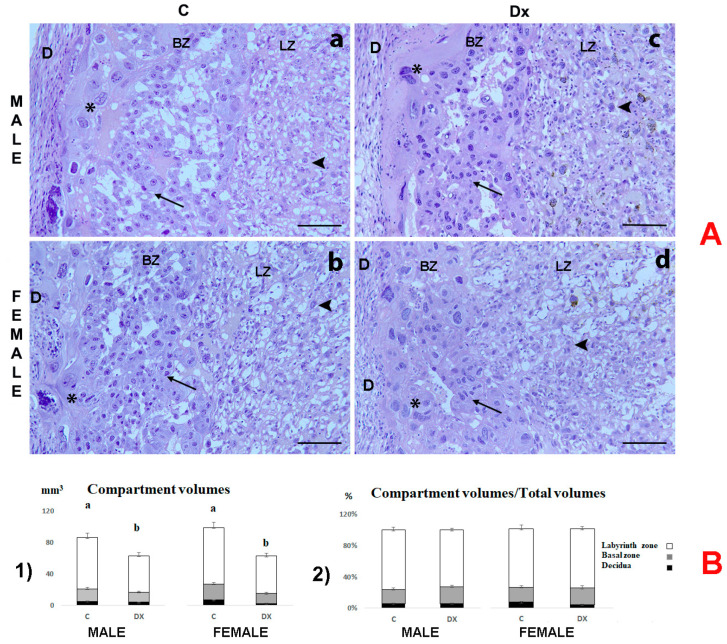
(**A**) Hematoxylin-stained sections of placentas from control (C; (**a**,**b**)) and dexamethasone-exposed (Dx; (**c**,**d**)) male and female fetuses on E 21. Decidua (D), basal zone (BZ), labyrinth zone (LZ), spongiotrophoblasts (arrows), giant cells (asterisk), trophoblast (arrowhead); scale bar = 100 μm; (**B**) (**1**) Absolute volume of placental zones (mm^3^); (**2**) Compartment zones density (%) for the placentas of male and female fetuses. Data are presented as mean ± SD, *n* = 6 placentas per group. Results were analyzed by two-way ANOVA. The groups not sharing a joint letter are significantly different (*p* < 0.05).

**Figure 3 ijms-24-00540-f003:**
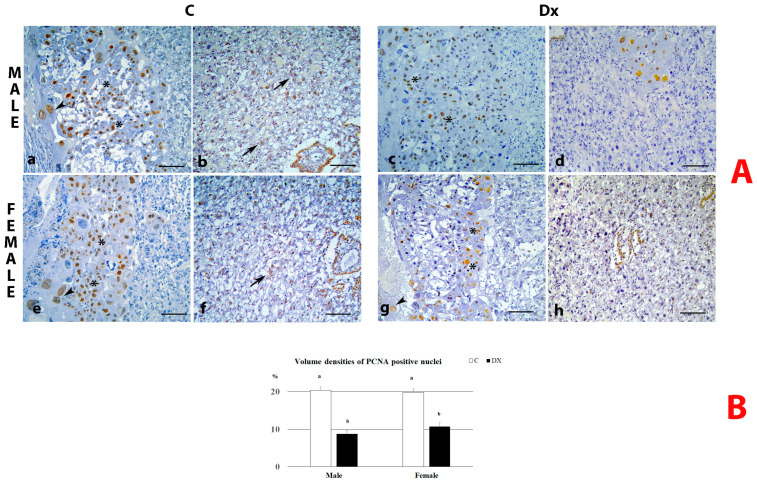
(**A**) PCNA immunoreactivity in placentas from control (C) and dexamethasone-exposed (Dx) male and female fetuses on E 21. PCNA immunostaining in basal (**a**,**c**,**e**,**g**) and labyrinth (**b**,**d**,**f**,**h**) zone. Spongiotrophoblasts (asterisk), giant cells (arrowhead), labyrinth trophoblasts (arrows); scale bar = 100 μm; (**B**) Volume densities of PCNA positive nuclei (%) for the placentas of male and female fetuses. Data are presented as mean ± SD, *n* = 6 placentas per group. Results were analyzed by two-way ANOVA. The groups not sharing a joint letter are significantly different (*p* < 0.05).

**Figure 4 ijms-24-00540-f004:**
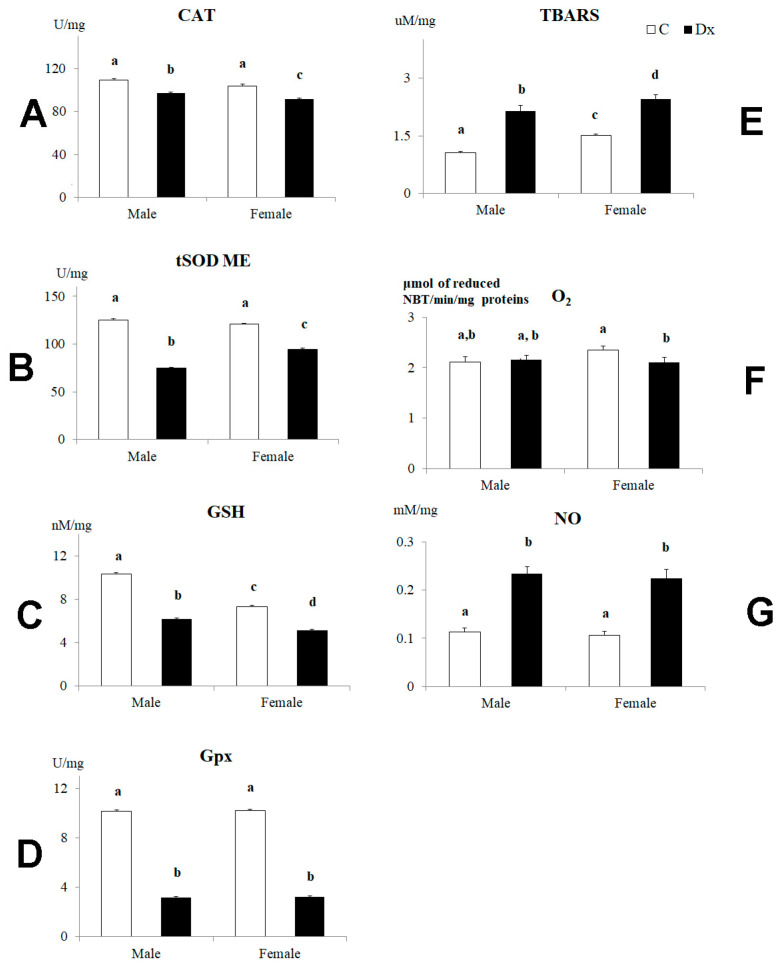
Antioxidative defense system in placentas from control (C) and dexamethasone-exposed (Dx) mothers of male and female fetuses on E 21. (**A**) Catalase (CAT, U/mg protein), (**B**) total superoxide dismutase (tSOD, U/mg protein), (**C**) glutathione (GSH, mU/mg protein) and (**D**) glutathione peroxidase (GPx, U/mg protein) activity; (**E**) levels of thiobarbituric acid reactive substances (TBARS, uM/mg), (**F**) superoxide anion radical (O_2_^•−^) and (**G**) nitric oxide (NO). Data are presented as mean ± SD, *n* = 6 placentas per group. Results were analyzed by two-way ANOVA. The groups not sharing a joint letter are significantly different (*p* < 0.05).

**Table 1 ijms-24-00540-t001:** Effects of maternal dexamethasone exposure on fetal weight, placental weight and placenta-to-fetal-weight ratio on E 21 of gestation.

	Male -Control (g ± SD)	Female-Control (g ± SD)	Male-Dx Group (g ± SD)	Female-Dx Group (g ± SD)
Fetal weight	5 ± 0.04	4.9 ± 0.03	4.2 ± 0.4 ^a,b^	4.1 ± 0.04 ^a,b^
Placental weight	0.62 ± 0.06	0.59 ± 0.05	0.36 ± 0.03 ^a,b^	0.35 ± 0.03 ^a,b^
Placental to fetal Weight ratio	0.124 ± 0.002	0.120 ± 0.003	0.082 ± 0.0001 ^a,b^	0.085 ± 0.003 ^a,b^

The results are mean ± SD, *n* = 15 fetuses or placentas (1–2 fetuses/placenta per one mothers’ litter) per group; ^a^
*p* < 0.05 vs. Dx-male, ^b^
*p* < 0.05 vs. Dx-female fetuses.

**Table 2 ijms-24-00540-t002:** Two-way factorial ANOVA testing of the effects of Dx treatment and fetal sex on CAT, tSOD, GSH, Gpx, TBARS, O_2_^•−^ and NO. * *p* < 0.001.

Factor	Treatment		Sex		Treatment × Sex	
	F (DFn, DFd)	P	F (DFn, DFd)	P	F (DFn, DFd)	P
CAT	682.3 (1, 24)	0.000 *	146.2 (1, 24)	0.000 *	0.0 (1, 24)	0.98
tSOD	7963.2 (1, 24)	0.000 *	315.3 (1, 24)	0.000 *	772.7 (1, 24)	0.000 *
GSH	5471.7 (1, 24)	0.000 *	2260.3 (1, 24)	0.000 *	514.1 (1, 24)	0.000 *
Gpx	34,768.1 (1, 24)	0.000 *	0.6 (1, 24)	0.444	0.0 (1, 24)	0.912
TBARS	662 (1, 24)	0.000 *	96.3 (1, 24)	0.000 *	3.3 (1, 24)	0.081
O_2_^•−^	8.5 (1, 24)	0.008	5.94 (1, 24)	0.023	17.8 (1, 24)	0.000 *
NO	479.3 (1, 24)	0.000 *	2.36 (1, 24)	0.136	0.1 (1, 24)	0.761

## Data Availability

Not applicable.

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
