# Peer review of "Maternal Dexamethasone Exposure Induces Sex-Specific Changes in Histomorphology and Redox Homeostasis of Rat Placenta"

_ijms, 2022, doi:10.3390/ijms24010540_

Round 1

Author Response

Reply to Reviewer 1

We appreciate the time and effort taken to review our manuscript, and wish to express gratitude for constructive comments and suggestions in order to improve its quality. We have carefully addressed all issues raised and made changes to the manuscript accordingly.

Reply to General comments:

  • Overall, manuscript grammar and writing is suitable for publication. No major issues when reading the manuscript, and any minor corrections are listed in the specific comments

-We thank the Reviewer for this comment.

  • The tables and figures are difficult to read in the manuscript file. For example, Table 1 is compressed and difficult to interpret without zooming in

- According to your suggestions, we made changes and increased the font size in all tables and figures.

  • Did the authors consider quantifying expression of antioxidant enzymes (whether it is gene expression of RNA transcripts or quantification of protein levels via a technique such as western blotting)? Theoretically, if Dex caused a change in gene or protein expression of antioxidant targets, without a change in efficiency/activity of the enzymes then the authors may not capture the complete picture of what is happening. Is there literature that has investigated such endpoints following Dex treatment? If so, this should be discussed as it is relevant to the activity results

- Gene and protein expressions of oxidative stress markers were not measured in this study. We agree with the Reviewer that changes in gene/protein expression with or without changes in enzyme activities may contribute to the results obtained in this study. Therefore, we included this mechanism together with the corresponding references to previously published results on changes in gene and protein expressions of antioxidative enzymes following maternal Dx exposure in the Discussion section (page 8, second paragraph).

  • Was maternal gestational weight gain quantified or compared across treatment groups?

-The maternal weight gain was monitored during the treatment and the results we present to the Reviewer in the graph and table below. The body mass of dams after the application of dexamethasone was slightly but not significantly reduced.

Upon your question, this result on final body weight of pregnant dams is added to Materials and Methods section (page 9, third paragraph).

Time point

C

0day

16day

21day

Dx

0 day

16 day

21 day

Average/group

262.8

306.8

347.2

273.7

316.7

327.5

SD

11.9

21.0

24.1

17.4

22.6

28.7

Reply to Specific Comments of Reviewer 1:

All the Specific Comments have been corrected:

  • Line 39: it would be helpful to briefly list the “long-term consequences” that the authors are eluding to.

-We have listed some of the long-term consequences such as altered behavior, metabolic and cardiovascular disease.

  • Line 64: what is the meaning of “superior maternal metabolism”? The wording is unclear if the authors mean that there is increased maternal metabolism during pregnancy? It would be helpful to clarify or use an alternate phrase possibly.

- Upon your suggestion, we have made the corresponding change in the text.

  • Table 1. “fetal weight” and “placental weight” in the left-most column should have first letters capitalized

-The Table 1. has been changed accordingly.

  • “Placental to fetal weight” in the left-most column is missing the term “ratio” at the end of the term

 -The Table 1. has been changed.

  • The caption indicates that it is n=15 fetuses or placentas per group, however how many dams do the 15 samples come from? This should be indicated

-The number of pregnant rats were 10 per group (control or Dx exposed).  1-2 fetuses/placenta per one mothers’ litter per group. The legend for Table 1. has been changed.

  • Figure 3: I would include the statistical test being used (2-way ANOVA) when indicating statistical significance using asterisk as figure captions should be stand-alone.

-We accepted your suggestion and made changes in the legends for figures accordingly.

  • Line 133: Text should read: “… exposure in the placentas of male and female fetuses respectively (Fig 4B)”

-We made corrections accordingly.

  • Figure 5

The notation for statistical significance differences is confusing to interpret when reading the figure. The distinction between the three symbols (asterisk, closed circle, open circle) in terms of which groups are being compared is not clear and confusing. …………….

-We accepted your suggestion and have made changes accordingly in the Figure 4 of revised manuscript.

  • Table 2: what are the degrees of freedom for the F-value in the ANOVA testing? Please provide appropriate notation for F-statistic (ex. Fdf, df = )

-The required values are added in the Table 2.

  • Line 223: “… and nonenzymatic vitamins C and E, zinc and copper, are present in the rat placenta”
    • o Did the authors purposely choose not to quantify or investigate these other antioxidant systems in the present study? Is there any information known in the literature regarding the responses of these antioxidants or other stressor models?
    •  

- Concentration of antioxidants such as vitamins C and E, and minerals zinc and copper, may also serve as a parameter of antioxidative defense.

Based on your question, it seems that we made confusion in the text and did not express what we actually meant with this general sentence. Therefore, as the information was not that relevant to the study, we removed this part of the sentence in the revised manuscript.

  • Line 225: at the end of this line, there is a sentence that begins with “And Maternal Dx….”. Please modify this sentence accordingly

-We made corrections accordingly.

  • Lines 248-250: there should be reference(s) to support this statement with literature

-The reference was actually provided at the beginning of the sentence, but we placed it now at its end upon your comment

  • Lines 252-255: it is beneficial to discuss the results of the present study to previous research from the same group due to replication of experimental conditions to support the findings, however the authors should also be comparing the results of this study to other rat or animal Dex treatment model studies. Although there would likely be differences in specific experimental conditions, it would be valuable to understand how the present findings fit into the current state of knowledge. The authors should add some text in the discussion to compare and contrast the present findings with other published Dex studies.

-Upon your suggestion, our data are placed in the context of other researchers’ results and the discussion has been extended (page 9, second paragraph).

  • Similar to the comment above, it would be valuable if the authors discussed how the present study and the results compare to nutritional models of programming. Is there evidence for sex-specific programming of offspring in nutritional models (ex. Caloric or protein restriction, overnutrition, micronutrient restriction, etc.), and if so, do the placental and redox outcomes parallel this study? It would be interesting to compare the findings of the present study to the known effects for nutritional programming models. Other stressors or diseases that can cause placental dysfunction (ex. Preeclampsia) could also be compared in the discussion. The current draft of the manuscript could do a better job discussing the context of the present study to the broader literature regarding placental response to IUGR or Dex.

- Based on the Reviewer’s comment we extended the discussion section together with corresponding references to include all suggestions in the text (page 8, fourth paragraph).

Reviewer 2 Report

In this manuscript, the authors demonstrated 1) dexamethasone reduced placental and fetal weight, 2) reduced placental zone volumes, 3) decreased the volume density of PCNA-immunopositive nuclei, 4) induced placental oxidative stress, and 5) some of these changes were prominent in the placentas of female fetus. Although these findings are interesting, no mechanistic insights are provided. The authors need to investigate whether dexamethasone-induced oxidative stress contribute to the reduced placental/fetal weight and reduced cell proliferation and how fetal sex plays a role in this regulation. The authors stated that ‘The concentration of superoxide anion radical (O2•-) was 11% higher (p<0.05) only in the placentas of female fetuses upon Dx exposure (Figure 5f)’ (lines 156-157). However, the bar graph of Figure 5f shows the opposite. In addition, the scale bar units in Figures 1, 2, and 4 (Lines 96, 107, and 138) should be correctly presented.

Author Response

Reply to Reviewer 2

We thank the Reviewer 2 for his time and effort to revise our manuscript. We also appreciate his useful comments and suggestions and have made changes in the manuscript accordingly.

In this manuscript, the authors demonstrated 1) dexamethasone reduced placental and fetal weight, 2) reduced placental zone volumes, 3) decreased the volume density of PCNA-immunopositive nuclei, 4) induced placental oxidative stress, and 5) some of these changes were prominent in the placentas of female fetus. Although these findings are interesting, no mechanistic insights are provided. The authors need to investigate whether dexamethasone-induced oxidative stress contribute to the reduced placental/fetal weight and reduced cell proliferation and how fetal sex plays a role in this regulation. The authors stated that ‘The concentration of superoxide anion radical (O2•-) was 11% higher (p<0.05) only in the placentas of female fetuses upon Dx exposure (Figure 5f)’ (lines 156-157). However, the bar graph of Figure 5f shows the opposite. In addition, the scale bar units in Figures 1, 2, and 4 (Lines 96, 107, and 138) should be correctly presented.

-Based on the Reviewer’s comment that ‘The authors need to investigate whether dexamethasone-induced oxidative stress contribute to the reduced placental/fetal weight and reduced cell proliferation and how fetal sex plays a role in this regulation’, it seems that we did not make our points clear.

We demonstrated that the maternal exposure to dexamethasone treatment reduced placental weight and the volume of all placenta zones. Reduced placental size means a reduction of the surface area for the transport of nutrients and oxygen to the fetus (Fowden et al. 2006), which under our experimental conditions probably caused decreased body weight of fetuses, This is in line with the results of other researchers who reported the positive correlation of placental size to body weight at birth term in various species (Fowden et al. 2006). Poor/excess nutrition, obstetric complications (preeclampsia, gestational diabetes, and hypertension) or abuse of toxic substances were also associated with decreased fetal growth, leading to low birth weight due to alterations in placental function (Myatt et al. 2016; Gaccioli et al. 2013; Biri et al. 2007; Schoots et al. 2018). Dexamethasone exposure was reported to decrease in the expression of genes involved in cell division in a murine late placenta (Baisden et al. 2007) and induction of oxidative stress (Jones et al. 2010), while oxidative stress is implied to contribute significantly to placental pathology (Schoots et al. 2018). Thus, both mechanisms that we examined, i.e. reduction of the proliferation rate, demonstrated as reduced number of PCNA-immunopositive nuclei, as well as increased oxidative stress, may directly contribute to examined effects of dexamethasone on reduction of placental size, its transport function, and consequently, on the lower fetal weight.

-Upon the Reviewer’s comment, we added this paragraph in the discussion section (page 9, lines 255-266) together with the corresponding references, and also made changes in the conclusion sections in the abstract section, as well as at the end of the text (page 9, lines 280-283).

-Based on the Reviewer’s comment ‘The authors stated that ‘The concentration of superoxide anion radical (O2•-) was 11% higher (p<0.05) only in the placentas of female fetuses upon Dx exposure (Figure 5f)’ (lines 156-157). However, the bar graph of Figure 5f shows the opposite.” we corrected the typing error in the manuscript text and changed it to 11% lower (page, line).

-Based on the Reviewer’s suggestion, the scale bar units in Figures 1, 2, and 4 (Lines 96, 107, and 138) are corrected.

Round 2

Reviewer 2 Report

The authors have addressed the reviewer's concerns.